# Modeled Cognitive Feedback to
# Calibrate Uncertainty for Interactive Learning

## Abstract

Many interactive learning environments use some measure of uncertainty to estimate how likely the model output is to be correct. The reliability of these estimates is diminished when changes in the environment cause incoming data to drift away from the data the model was trained on. While interactive learning approaches can use implicit feedback to help tune machine learning models to identify and respond to concept drift more quickly, this approach still requires waiting for user feedback before the problem of concept drift can be addressed. We propose that modeled cognitive feedback can supplement implicit feedback by providing human-tuned features to train an uncertainty model that is more resilient to concept drift. In this paper, we introduce modeled cognitive feedback to support interactive learning, and show that an uncertainty model with cognitive features performs better than a baseline model in an environment with concept drift.

## 1. Introduction

Machine learning (ML) models operating in dynamic real world environments often experience degraded performance as the incoming data changes from the training set. If the concept drift is not addressed, it can lead to incorrect model-based decisions. Adjusting for such changes in traditional ML systems usually requires monitoring the model performance and then completing slow and costly retraining when the performance falls below some threshold. Interactive learning opens up new opportunities for refining machine learning models in the face of concept drift, by training and improving models online through implicit feedback collected from user activity. However, this process relies on waiting for user feedback before the model can begin to

[1]Anonymous Institution, Anonymous City, Anonymous Region, Anonymous Country. Correspondence to: Anonymous Author <anon.email@domain.com>.

Preliminary work. Under review by the International Conference on Machine Learning (ICML). Do not distribute.

improve, and this can often take more time and interaction than desired. When a system is slow to detect and respond to concept drift, it will be incorrect in its estimates of uncertainty. This can be especially problematic in workflows where an analyst is working to validate and correct machine generated labels. In this situation, uncertainty models are used to prioritize what labels are shown to the analyst without overwhelming them with too many incorrect guesses (Michael et al., 2020). This process relies on an uncertainty model that accurately estimates the probability of being wrong about a classification. If the model underestimates or overestimates the probability of a particular label, this can degrade the performance of the human-machine team.

In this paper, we investigate using modeled cognitive feedback to supplement user feedback to tune an uncertainty model to more accurately reflect the data distribution in a changing environment. First, we will introduce the challenges of representing uncertainty in interactive learning environments. We will then provide some background on cognitive models and how they have been used to support both interactive and machine learning systems, and discuss how they could be used to more quickly adapt to concept drift in interactive learning environments. Finally, we will describe a proof of concept where we compare two uncertainty models in an online learning task and show that one incorporating cognitive features derived from modeled visual search is more accurate over time than one using more traditional features.

## 2. Background

### 2.1. Uncertainty in Interactive Learning

We consider the challenge of representing uncertainty in an interactive learning system where an analyst is working closely with a machine learning system to validate and correct labels. Uncertainty is often not well-defined, but it is often assumed to be some measure of what is unknown (Weinhardt & Schaefer, 2022) Here we define it as the probability of machine correctness. This measure is used in active learning to identify examples that would be the most impactful in training a supervised model, minimizing the number of examples that the analyst must label or validate

to help the model converge (Cohn et al., 1994) . When an uncertainty model is inaccurate, it leads to a model that is either underconfident or overconfident. An overconfident model may make more mistakes by not prioritizing the validation of labels that are wrong, while an underconfident model may waste resources asking for validation when none is needed. Calibrating an uncertainty model in a dynamic real world environment calls for interactive tools that build upon both the human and machine strengths to identify and adapt to different kinds of change in the data. For example, automated methods can be used to adjust the uncertainty model when performance metrics fall below a threshold (Bayram et al., 2022), but relying on this method alone is slow to detect problems. Interactive learning environments offer alternative ways to detect concept drift by using explicit and implicit feedback collected from a user who may identify changes in the environment or a drop in machine classification accuracy before the automated methods. However, waiting for user feedback is still not ideal, since the uncertainty model is likely already inaccurate by the time the user can identify the problem and provide feedback.

In the remainder of the paper, we will introduce a new approach for providing cognitive feedback to calibrate an uncertainty model. This approach builds on previous work using cognitive models to design adaptive interfaces and machine learning models that must be built upon some understanding of human behavior. We will start with describing some of the previous work done to incorporate cognitive models into human-attuned interfaces, and then provide an example introducing how cognitive models can be used to generate cognitive features that can help an uncertainty model be more resilient to concept drift in an interactive task.

## 2.2. Cognitive Modeling for Human-Attuned Interfaces

Cognitive models are built upon clearly defined theories about aspects of cognition, such as memory, learning or attention. They provide an algorithmic representation of a psychological theory that simulates a measurable aspect of human performance (i.e. reaction time, accuracy). The resulting simulation can be compared to real human performance to validate the model's strengths and weaknesses (Lewandowsky & Farrell, 2010). Many cognitive models leverage existing cognitive architectures in their design. Cognitive architectures represent a specific theory about how human minds are structured, allowing them to learn, reason, and/or perceive the environment. These, too, have been developed through systematic evaluation against human performance in psychological studies. Many cognitive architectures exist, each with their own design choices. For example, SOAR is a cognitive architecture that incorporates several modules that run in parallel and are controlled by a procedural rule-based system. It differentiates between

working, episodic, and semantic memory and incorporates visuospatial and motor modules to control virtual effectors (Laird et al., 1986). ACT-R is another architecture that incorporates modules that represent a variety of aspects of cognition, including declarative memory, visual attention, auditory attention and motor functions. These modules can run in parallel around a central, rule-based control system (Anderson et al., 2004). Many other architectures exist beyond these two, each with their own strengths and weaknesses. Recent work has considered how to unify these into a common computational framework that represents theory where architectures generally agree (Laird et al., 2017). By building upon cognitive architectures that have been validated against human performance, a cognitive model can provide a hypothesis as to how humans would respond to specific tasks involving the cognitive functions modeled by that architecture. Cognitive models support the design of human-attuned interfaces by providing a baseline algorithmic representation of human cognitive abilities and limitations.

In the following sections, we will explore how research in human factors and machine learning has previously leveraged cognitive models to create human-attuned interfaces and models. We will then consider the potential for cognitive models in providing feedback for interactive learning. Finally, we will introduce the challenges in representing uncertainty in interactive learning workflows and consider how modeled cognitive feedback can help machine learning models respond more rapidly to concept drift.

### 2.2.1. COGNITIVE MODELS IN HUMAN FACTORS RESEARCH

Human Factors research has a long history of incorporating cognitive models. Often this is done to help define a specific theory that can explain some observed aspect of human performance and test how changes to the interface or environment might affect performance. In turn, this can be used to improve the system or interface that a user interacts with. In Salvucci (2006), researchers developed a model of a car driver in ACT-R. The model could account for the steering behavior and gaze distributions of human drivers in a multilane highway environment. The work provided an initial example of applying models designed in ACT-R to complex, realistic tasks. In another example, Fleetwood & Byrne (2006) developed an ACT-R model of visual search to describe how participants of an eye tracking study searched the screen for an icon. The model was used to explain both response time and eye movement data. In another study, researchers used psychological theories and eye tracking data to develop an ACT-R model to simulate the relative difficulty in recognizing different messages in grouped bar charts (Burns et al., 2013). In Lohrenz et al. (2009), cognitive models of clutter were developed to ex-

plain people's subjective ratings of clutter on geospatial displays. This model was used to help evaluate whether a geospatial display would be considered too cluttered by its intended audience. Cognitive models have also been developed to model visual search patterns and timings for familiar layout designs. By incorporating learning and memory into the model, it was possible to predict when layouts become familiar or forgotten and predict how much a familiar layout might impact a user's visual search behavior when exploring a new unfamiliar layout (Todi et al., 2018).

### 2.2.2. COGNITIVE MODELS IN MACHINE LEARNING RESEARCH

Machine learning has also benefited from incorporating features and simulated data from cognitive models. For example, Plonsky et al. (2016) extended a random forest algorithm to include psychological features in addition to more standard and naive features. In a choice prediction competition, the resulting model significantly outperformed other models built upon best practices. To address the challenge of predicting human decisions, which often require huge datasets to accurately model with off the shelf techniques, Bourgin et al. (2019) generated data from cognitive models of decision making and used these to pretrain a neural network. The network was later finetuned using a smaller sample of real human decision making data. This approach led to improvements on two benchmark datasets. (Trafton et al., 2020) also explored generating synthetic data to support machine learning models of human behavior. This research explored using ACT-R models explaining different strategies of behavior in a supervisory control task to generate synthetic data to supplement real human data when training a convolutional network. The best results were achieved by combining real human data with synthetic data generated from the different strategies, which performed better than models trained off empirical or synthetic data alone.

### 2.3. Cognitive Models for Interactive Learning

We have reviewed several examples of how cognitive models can simulate human cognitive abilities to help make human factors decisions. We also explored past research showing how cognitive models can reduce the amount of real human data required to create machine learning models with equivalent or better performance of those trained on human data alone. Modeled cognitive feedback for interactive learning could build upon this work by simulating cognitive aspects of interfacing with an interactive learning system and looking at measurable metrics, such as fixation locations in a visual search or the reaction time to find and select a button. This information can be incorporated as an additional feature into the machine learning model, or as feedback to a reinforcement learning algorithm. Changes

in modeled reactions could be an indication that something about the underlying environment has also changed so that the uncertainty model can adapt even before an active learning algorithm selects a data point to query the user about.

We will now introduce a task designed to compare uncertainty models in an interactive learning environment with concept drift.

## 3. Threshold Selection Task

We designed a simple threshold selection task to explore methods of evaluating uncertainty as a probability of machine correctness. The goal of the task is to locate the threshold on a noisy signal graph where the signal no longer appears to be high. The signal graph is designed as a sigmoid curve with varying degrees of noise, and the user can select any point along the curve that they think is the point of inflection. This task was designed to provide an intuitive problem where the machine learning algorithm predicts the threshold location that a user would choose. Over the course of several examples, the machine learning model guesses where the threshold will be and then the user clicks the point they think best represents the threshold.

Throughout the task, the user is presented with graphs generated from 5 sets of signal types. Each signal type incorporate noise into the sigmoid differently to introduce concept drift (see Figure 1 for examples). Each of the 5 sets contains 7 trials where noise is generated in that same way. 30 realizations, each representing a user and machine team stepping through 35 trials, are completed. By collecting data over 30 realizations, we can consider the distribution of machine and user placements to calculate the probability of machine correctness and compare it to the confidence score produced by the uncertainty model.

We will now describe the classifier used to select the machine placements from which the uncertainty models predict a confidence value (probability of correctness).

### 3.1. Classifier

A naïve Bayes classifier predicts the user placements using several features chosen from standard feature selection techniques. To train the classifier, labels were determined by assigning 1 to all sample points with x-coordinates less than or equal to the human placed threshold and 0 to the remaining points.

### 3.2. Cognitive Model

In addition to generating machine placements, we designed a cognitive model to generate cognitive features for each trial that were provided to the cognitive uncertainty model. The model simulates a user visually scanning and encoding

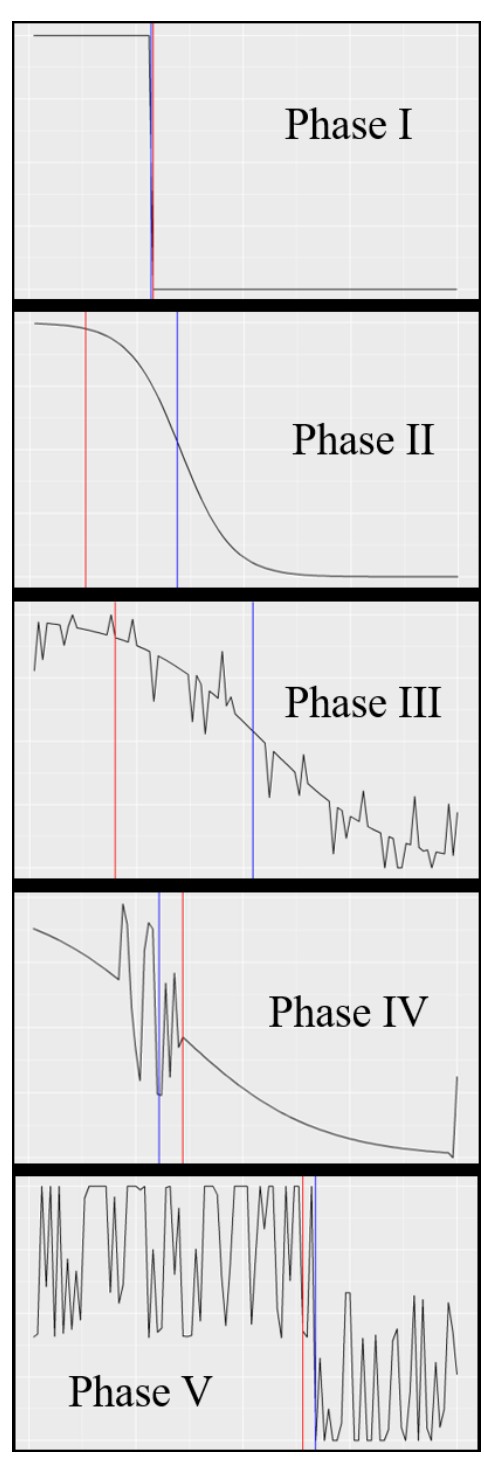

*Figure 1.* Exemplar graphs from the five different phases from the threshold selection task

a sigmoid curve. We designed the cognitive model in ACT-R, using the EMMA extension. EMMA extends ACT-R to include basic functionality to generate quantitative predictions about eye movements, including the timing of those movements (Salvucci, 2001). The ACT-R agent is designed to simulate the eye movements that occur as a user scans along the sigmoid curve and selecting a specific point as the inflection point. Since the location of the inflection point is subjective, depending on a user's preferences, our ACT-R agent does not simulate the decision of choosing a point and instead chooses one at random. From the simulation, we were able to extract timing information about the task, including the total amount of time spent scanning the curve to points that were fixated upon long enough to be encoded. We used this information to design three cognitive features for our uncertainty model. These are defined below and were generated for each x-coordinate along the sigmoid.

- **scan-time** ($t$): The amount of time that passed between trial start and a fixation point along the sigmoid curve, as calculated by EMMA. The number of points that were fixated upon, and therefore the number of total points along the line with associated scan-time values, varied depending on the shape and noise level of the sigmoid. All other scan-time values are set to 0.

- **extrapolated-time** ($t_e$): A timing calculated from the scan-time that assigns an extrapolated-time to every x-coordinate position $pos(x)$ between the two fixation positions ($pos(p_0)$ and $pos(p_1)$). The extrapolated time $t_e$ can be calculated as:

$$t_e = t(p_0) + \frac{(pos(x) - pos(p_0))(t(p_1) - t(p_0))}{pos(p_1) - pos(p_0)}$$

- **next-time** ($t_1$): Each points $t_1$ value is set to the scan-time ($t$) for the associated x-coordinate if it exists. Otherwise, it is set to $t$ associated with the next lowest x-coordinate that has a scan-time value associated with it.

### 3.3. Uncertainty Model

The cognitive model described above provides three features that can be provided to an uncertainty model when it's deciding about how confident it is in the classifiers decision. We define uncertainty to be the probability of machine incorrectness for a given placement. The uncertainty value can be used to report machine confidence, which is defined as $confidence = 1 - uncertainty$

*Baseline Uncertainty Model* To examine the effectiveness of using cognitive features in our uncertainty model, we compare two versions of the uncertainty model. The first version is a baseline uncertainty model. This model also naïve Bayes, and draws heavily from the classifier model.

However, it differs from the classifier in that the uncertainty model allows some placement tolerance such that any point within some distance of a user placement is considered a correct placement. Lower placement tolerances should lead to lower expected accuracy. Points within the tolerance distance are labeled as 1 and the rest are labeled as 0.

*Cognitive Uncertainty Model* The second version of the uncertainty model is built in the same way as the baseline uncertainty model, except now we are using three additional cognitive features (scan-time, extrapolated-time, and next-time) derived from the line scan model described above in Section 3.2.

## 4. Results

The two uncertainty models described above were used to generate confidence scores for the machine placements generated from the classifier. We evaluated the confidence scores by comparing them to the probability of machine correctness using the mean absolute error. By comparing the mean absolute error of the baseline uncertainty model to that of the cognitive uncertainty model, we see that while the models are comparable in early trials, the cognitive uncertainty model trends towards being a better predictor of machine correctness over time (see Figure 2).

Recall that both uncertainty models made use of a tolerance parameter that configures how close a machine placement needs to be to the user's chosen inflection point to be considered correct. When we consider the average mean absolute error of both uncertainty models at different tolerances, we see the same trend, with the cognitive uncertainty model outperforming the baseline uncertainty model in the latter half of the trials (see Figure 2(c)).

## 5. Conclusions and Future Work

We have discussed the importance of accurate uncertainty models in interactive learning environments and the challenges of calibrating uncertainty models to account for concept drift as real world data evolves away from the data a model was initially trained on. A new method was introduced to use modeled cognitive feedback to improve uncertainty models, even before feedback is collected from the user. An interactive learning task was designed to explore the potential of this method. In the task, a user and a machine learning model work together to select the inflection point of as noisy sigmoid curve. A cognitive uncertainty model was used to generate confidence values for the machine learning placements, and it incorporated three features derived from a cognitive model of a user visually scanning the sigmoid graph in each trial. When compared to a baseline uncertainty model, we found that the cognitive uncertainty model trended towards predicting confidence scores

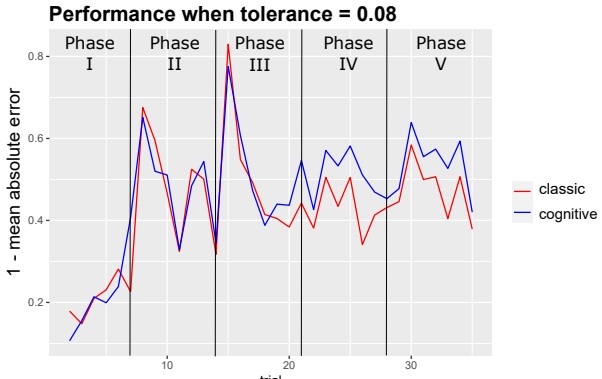

(a) Perfomance of uncertainty models across trials when tolerance is 0.08.

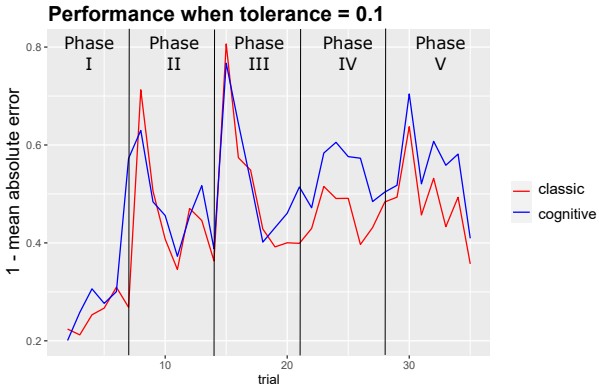

(b) Perfomance of uncertainty models across trials when tolerance is 0.1

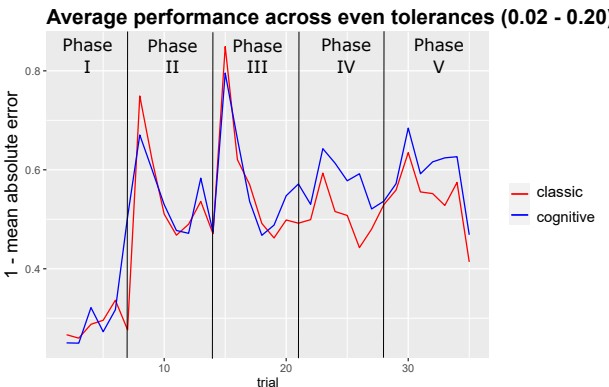

(c) Average performance of uncertainty models across trials for all tolerances explored (even values between 0.2-0.20)

*Figure 2.* The performance of the uncertainty models using cognitive features is comparable to the model using classic features in early trials and consistently higher in later trials.

that were a closer representation of machine correctness than the baseline model in later trials. This seems to indicate that the cognitive uncertainty model was more resilient to the changes in the signal data as it became more noisy, but required some trials to train on the cognitive features.

This represents an initial result that cognitive features derived from a cognitive model can improve performance of underlying uncertainty models compared to a baseline model. There is still room for improvement upon this method. Currently, the cognitive model that generates the features is designed using a single visual search strategy. It is possible that a more robust cognitive model could be developed by analyzing eye tracking data from users completing the task. It would be interesting to see how much the performance of the cognitive uncertainty model could be improved by using features from a more realistic cognitive model.

In this paper, we considered interactive threshold detection as a simple task to test if cognitive features could help the uncertainty model performance. In the future, we would like to explore developing cognitive uncertainty models for realistic interactive learning tasks, such as region digitization or sound identification.

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
