# OpenReview forum: "Modeled Cognitive Feedback to Calibrate Uncertainty for Interactive Learning"
_ICML.cc/2023/Workshop/ILHF — ILHF Workshop ICML 2023_

### Official Review · Reviewer_rZ76 · 2023-06-14

**Rating:** 4
**Confidence:** 2

**Review:**

This paper addresses the problem of how to build machine learning models that can respond when there is a shift in data distribution given to a machine learning model, or “concept drift”. The authors propose that modeling human cognitive feedback supplements feedback from humans about where concept drift can occur. The authors propose a task of locating the threshold on a signal graph where it no longer appears to be high. They compare a naive Bayes classifier to a model that in addition uses features generated from an ACT-R simulation.

I am not an expert on cognitive modeling or interactive learning.

Strengths:
- The problem of locating concept drift is an important one towards being able to deploy machine learning systems in the real world, and human feedback is a promising avenue towards improving performance on these tasks.
- The paper does a good job of introducing research on cognitive models for human factors research, and is generally well-written.

Weaknesses:
- It is unclear if the results are statistically significant – no statistical tests have been conducted to put the results for example from Figure 2 into context. As a result, the claim in the caption in Figure 2 doesn’t seem to be fully justified.
- It seems that while the uncertainty model introduces human-cognitive features, these features may be computed as statistics of the inputs given to the naive model. Perhaps it would be convincing to explore if the same results are seen in a setting with a more sophisticated model.
- There aren’t any comparisons to methods from prior works, including simple methods such as those described from Bayram et al. 2022 that adjust the uncertainty when metrics fall below a certain threshold. The task also appears to be newly introduced in this paper, so it is difficult to determine how strong the results are.

---

### Official Review · Reviewer_RUBs · 2023-06-19
**Simple Experimental Setup and Promising First Results**

**Rating:** 6
**Confidence:** 3

**Review:**

## Summary
- Background
	- Concept drift affects the performance of deployed machine learning models.
	- Interactive learning addresses this problem via human feedback but requires well-calibrated uncertainty estimates of the system's output.
	- Cognitive models take the psychology of humans interacting with the systems into account and model attention, memory and motor functions.
- Method
	- Design a threshold selection task where the users and models select the point of inflection in a sigmoid signal with five different noise patterns.
	- Create cognitive model of the user in ACT-R + EMMA and extract eye-movement timing features for the uncertainty model.
	- Compare naive Bayes classifier with and without cognitive features in predicting the points of inflection.
- Experiments
	- 30 users, each with 7 iterations of the same noise process label the points in a sequential setup.
	- Mean absolute error is evaluated for the augmented classifier and the baseline for different thresholds.

## Strengths
- Provides a good introduction into the problem setting and related background.
- Simple setup to explore the use of cognitive models for uncertainty estimation.

## Points of Improvement
- The caption in Figure 1 should explain the red and blue threshold.
- A table listing the correctness or mean absolute error for each of the phases together with their standard deviations would be helpful. Figure 2 shows the advantage but numerical results leave less room for interpretation.
- As the setup is quite minimal, the feature importance for the models would also be interesting to evaluate. We see that the naive Bayes with cognitive features works better but which feature attributes most to the class labels is an interesting point in my view.

---

### Decision · Program_Chairs · 2023-06-20

Accept